# LKB1 Loss Correlates with STING Loss and, in Cooperation with β-Catenin Membranous Loss, Indicates Poor Prognosis in Patients with Operable Non-Small Cell Lung Cancer

**DOI:** 10.3390/cancers16101818

**Published:** 2024-05-10

**Authors:** Eleni D. Lagoudaki, Anastasios V. Koutsopoulos, Maria Sfakianaki, Chara Papadaki, Georgios C. Manikis, Alexandra Voutsina, Maria Trypaki, Eleftheria Tsakalaki, Georgia Fiolitaki, Dora Hatzidaki, Emmanuel Yiachnakis, Dimitra Koumaki, Dimitrios Mavroudis, Maria Tzardi, Efstathios N. Stathopoulos, Kostas Marias, Vassilis Georgoulias, John Souglakos

**Affiliations:** 1Department of Pathology, University General Hospital of Heraklion, 71500 Heraklion, Greece; akoutsop@uoc.gr (A.V.K.); tzardi@med.uoc.gr (M.T.); stath@med.uoc.gr (E.N.S.); 2School of Medicine, University of Crete, 70013 Heraklion, Greece; mavroudis@uoc.gr (D.M.); georgoul@uoc.gr (V.G.); j.souglakos@med.uoc.gr (J.S.); 3Laboratory of Translational Oncology, School of Medicine, University of Crete, 70013 Heraklion, Greece; msfakianak@uoc.gr (M.S.); chapapadak@uoc.gr (C.P.); voutsina@med.uoc.gr (A.V.); chemp945@edu.chemistry.uoc.gr (M.T.); med4p1030249@med.uoc.gr (E.T.); fiolitakigeorgia@pagni.gr (G.F.); dorachat@med.uoc.gr (D.H.); 4Foundation for Research and Technology Hellas (FORTH), 70013 Heraklion, Greece; gmanikis@ics.forth.gr (G.C.M.); kmarias@ics.forth.gr (K.M.); 5Laboratory of Bio-Medical Data Analysis Digital Applications and Interdisciplinary Approaches, University of Crete, 71003 Heraklion, Greece; giahnaki@med.uoc.gr; 6Department of Dermatology, University General Hospital of Heraklion, Voutes, 71500 Heraklion, Greece; dkoumaki@pagni.gr; 7Department of Medical Oncology, University General Hospital of Heraklion, 71500 Heraklion, Greece

**Keywords:** NSCLC, LKB1, STING, PD-L1, β-catenin, pAMPK, ΚRAS, BRAF, VEGFC, PDGFRβ, ZEB-1, Cyclin D1, p16

## Abstract

**Simple Summary:**

Deleterious LKB1 inactivation in KRAS-driven lung adenocarcinomas (LUACs) has been studied thoroughly; however, LKB1 inactivation is not limited in adenocarcinomatous histology, nor in KRAS mutational status. We investigated LKB1 loss by IHC in 188 metastatic and 60 non-metastatic tumors from patients with operable non-small cell lung cancer (NSCLC) of various histologies and its correlation with cell cycle targets, EMT inducers, antitumor immunity response regulators, cell adhesion molecules and KRAS co-mutations. LKB1 loss was associated with STING loss irrespective of KRAS status both overall and in lung adenocarcinomas (LUACs) and, importantly, also in the metastatic setting. Concurrent LKB1 loss and β-catenin loss of membranous expression correlated with a significant decrease in median overall survival and increased risk of death. The expression status of LKB1 in association with STING and β-catenin membranous expression status could be used as stratification factors for patients with operable NSCLC in the procedure of precision medicine in lung cancer.

**Abstract:**

To investigate the incidence and prognostically significant correlations and cooperations of LKB1 loss of expression in non-small cell lung cancer (NSCLC), surgical specimens from 188 metastatic and 60 non-metastatic operable stage I-IIIA NSCLC patients were analyzed to evaluate their expression of LKB1 and pAMPK proteins in relation to various processes. The investigated factors included antitumor immunity response regulators STING and PD-L1; pro-angiogenic, EMT and cell cycle targets, as well as metastasis-related (VEGFC, PDGFRα, PDGFRβ, p53, p16, Cyclin D1, ZEB1, CD24) targets; and cell adhesion (β-catenin) molecules. The protein expression levels were evaluated via immunohistochemistry; the RNA levels of LKB1 and NEDD9 were evaluated via PCR, while KRAS exon 2 and BRAF^V600E^ mutations were evaluated by Sanger sequencing. Overall, loss of LKB1 protein expression was observed in 21% (51/248) patients and correlated significantly with histotype (*p* < 0.001), KRAS mutations (*p* < 0.001), KC status (concomitant KRAS mutation and p16 downregulation) (*p* < 0.001), STING loss (*p* < 0.001), and high CD24 expression (*p* < 0.001). STING loss also correlated significantly with loss of LKB1 expression in the metastatic setting both overall (*p* = 0.014) and in lung adenocarcinomas (LUACs) (*p* = 0.005). Additionally, LKB1 loss correlated significantly with a lack of or low β-catenin membranous expression exclusively in LUACs, both independently of the metastatic status (*p* = 0.019) and in the metastatic setting (*p* = 0.007). Patients with tumors yielding LKB1 loss and concomitant nonexistent or low β-catenin membrane expression experienced significantly inferior median overall survival of 20.50 vs. 52.99 months; *p* < 0.001 as well as significantly greater risk of death (HR: 3.32, 95% c.i.: 1.71–6.43; *p* <0.001). Our findings underscore the impact of the synergy of LKB1 with STING and β-catenin in NSCLC, in prognosis.

## 1. Introduction

Lung cancer remains the leading cause of cancer-related death worldwide, estimated to be responsible for one in four deaths (25.3%) [1,2]. The stage at diagnosis determines treatment options and has the strongest influence on survival in NSCLC patients. Approximately half of the patients undergoing complete surgical resection for low-stage disease die from recurrent disease or distant metastases.

The STK11/LKB1 gene was identified when germline heterozygous mutations in the p13.3 region of chromosome 19 were revealed as the causal mutation in Peutz–Jeghers Syndrome (PJS), an autosomal dominant disorder characterized by the development of benign gastrointestinal hamartomas, and were associated with the early onset of cancer [3,4]. LKB1 gene codes for Liver Kinase B1 (LKB1) an evolutionarily conserved serine threonine kinase with manifold tasks and master upstream kinase of AMPK, a major regulator of energy sensing and cellular metabolism [5,6,7]. Moreover, LKB1 is a master kinase of at least 13 more AMPK-related kinases and is in a constitutively active state in growing cells [8].

The predisposition of patients with PJS to early onset tumors indicates that LKB1 inactivation is an important event in carcinogenesis. Genetic analysis acknowledged LKB1 as a critical tumor-suppressor gene that is frequently inactivated in a broad spectrum of human cancers. LKB1 inactivation takes place mostly through somatic alterations in the LKB1 gene, such as nonsense mutation, loss of heterozygosity, insertions, intragenic deletions, or chromosomal deletions, the majority of which affect the kinase domain, resulting in a loss of kinase activity, while others affect the production, stability, or localization of the protein [9,10,11,12,13,14,15,16,17]. Additionally, alterations in LKB1 expression can also occur as a result of non-mutational mechanisms [18] such as hyper-methylation of the promoter region [19,20], as well as at the post-translational level, at which they are induced by micro-RNAs [21].

In a study examining 4446 patients with various solid tumors, the rate of LKB1 alterations in the pan-cancer setting was 1.35%, the overwhelming majority (45%) of which were NSCLC-related [22]. Remarkably, LKB1 was found to be the third most commonly altered gene in LUACs, after TP53 and KRAS [23]. LKB1 inactivation occurs early in the development of lung cancer, as demonstrated by the loss of LKB1 protein expression in atypical adenomatous hyperplasia (AAH) [24] and in LUACs in early pathological stages (I and II) [25].

The most frequently LKB1 co-mutated genes are KRAS, TP53, BRAF, and CDKN2A(HD), with KRAS and TP53 co-mutation being present in about half of LKB1-altered cancers [23,26,27]. The prevalence of LKB1 alterations in LUACs is 8–21% [27,28,29,30,31], whereas in Lung Squamous Cell Carcinoma (LSCCs), LKB1 alterations are detected in 1.5–5% of patients [26,28]. Various studies have highlighted the role of LKB1 loss as an enhancer of cell growth, motility, invasion, and as a metastasis promoter.

In a seminal paper, Ji et al. used a mutant KRAS-driven model of mouse lung cancer to show that LKB1 inactivation stimulated the growth of tumors of various histotypes with vigorously strong shortened latency to growth and to metastasis development compared to the LKB1-wild-type (WT) counterpart. Furthermore, LKB1-deficient tumors yielded increased expression of genes involved in angiogenesis and cell migration, such as NEDD9 and CD24 [32].

In vitro studies revealed that LKB1 inactivation increased cell motility and invasiveness [16,33] and favored epithelial–mesenchymal transition in lung cancer cells through upregulation of Zinc-finger E-box-binding homeobox factor 1 (ZEB1), a transcriptional repressor for E-cadherin and an EMT inducer, thus enhancing metastatic potential [34,35].

Moreover, in recent years, increasing amounts of evidence have highlighted the role of LKB1 in modulating the tumor immune microenvironment. Skoulidis et al., focusing on KRAS mutant LUACs, revealed different co-mutational subsets of KRAS mutants with distinct biologic and vulnerabilities and immune profiles [36] and showed that KL tumors exhibited lower densities of CD3+ and CD8+ lymphocytes, but not FOXP3+ cells, and lacked PD-L1 expression in the tumor cells despite intermediate–high TMB [37].

Nearly half of the KRAS-mutant tumors harbor concomitant TP53 mutations, followed by LKB1 (18% to 28%) and KEAP1 (24%), while less than 5% of KRAS-mutant NSCLC patients have another oncogenic-driven co-mutation, such as BRAF, EGFR, PIK3CA, or MET amplification, with no concurrent ALK or ROS1 rearrangements described [27,28,29,36,37,38].

Considering all of the above factors, we performed a comprehensive retrospective study in a large series of patients with resectable (stage I-IIIA) NSCLC with the aim of performing an in situ immunohistochemical evaluation of the incidence of loss of LKB1 expression in surgical specimens of NSCLC patients with and without lymph node metastasis, including all of the various histotypes, in order to elucidate possible patterns of expression and characteristics of the LKBl-less phenotype and to identify potential correlations of prognostic significance.

Furthermore, we investigated the incidence and the characteristics of tumors characterized by concurrent KRAS mutations with loss of LKB1 expression (now referred to as KL), with p53 mutational immunohistochemical status (now referred to as KP), and with p16 downregulation (hereafter referred to as KC), the KRAS mutant and LKB1 intact combination, as well as triple mutants KRAS-p53-p16 (hereafter referred to as KPL), LKB-only mutants (hereafter referred to as L status), which are defined as tumors with LKB1 loss and no KRAS, p53, p16 co-mutations, and of KRAS-only tumors with no identifiable LKB1, p53, and p16 commutations (hereafter referred to as K status) and their impact on prognosis.

## 2. Materials and Methods

### 2.1. Patient Population/Study Design

To perform this retrospective translational research study, we searched the records of the Pathology department of the University General Hospital of Heraklion and retrieved the pathology reports of patients who underwent surgery, with curative intent, for NSCLC between 2007 and 2023. Successively, and after re-evaluating the slides of excised tumor tissue and excised lymph node specimens slides for each patient to determine (i) the histologic subtype according to the 2015 WHO classification [39] and (ii) the stage in accordance with the 8th American Joint Committee on Cancer Staging AJCC [40], we enrolled 188 treatment-naïve patients with operable, stage I-IIIA NSCLC harboring pathologically confirmed metastasis of regional and distant lymph nodes, whom we assigned to the experimental group. Meanwhile, 60 treatment-naive patients with pathologically confirmed tumor-free lymph nodes were assigned to the control group. Furthermore, patient selection for enrollment did not account for the histological type and mutational status of the tumor to ensure that the entire repertoire of histologies of NSCLCs and all the potential mutational cohorts were included. 

The clinical data of the patients were collected from the medical records of the Department of Medical Oncology, University General Hospital of Heraklion. The study was conducted in accordance with the Declaration of Helsinki and Good Clinical Practice guidelines and after approval by the Ethics Committee of the Institution. Prior to surgery, all patients gave their informed written consent for the use of tumor samples for laboratory analysis. All laboratory analyses were performed blinded to clinical data.

### 2.2. Specimen’s Characteristics and Assays

Four-micron-thick formalin-fixed, paraffin-embedded (FFPE) tissue sections were prepared on charged glass slides as per the standard protocols for IHC. Tissue sections were deparaffinized and peroxidase was quenched with methanol and 3% H2O2 for 15 min. Heat-induced epitope retrieval was performed by immersing the slides either in EDTA buffer at pH8 or in Citrate buffer at pH6, followed by two cycles of microwave treatment at 500 watts. After a thorough rinse with distilled water, primary antibodies were applied at validated dilutions and incubation times (Appendix A). UltravisionTM Quanto Detection System HRP (Thermo Scientific™, Waltham, MA, USA) was used to detect antibody binding, according to the manufacturer’s recommendations. Slides were counterstained with hematoxylin PD-L1 and expression status was assessed using the VENTANA PD-L1 (SP142) assay (ROCHE; Basel, Switzerland).

Immunohistochemical evaluation of the LKB1 [37,41], pAMPK [42], STING [43], ZEB1 [34,35], p53 [44], Cyclin D1, p16 [45,46,47], PDGFRα, PDGFRβ [48], CD24 [49], VEGFC [50], and β-catenin [51] expression status was performed by two independent pathologists (E.L. and A.K.) based on previously published data. A detailed description of the methodology [Appendix A] and a table of the antibodies used [Appendix A] are provided. In all experiments, along with tumor tissue, matched adjacent normal lung tissue that was located at least 5 cm away from the tumor lesions was selected and evaluated. Regarding LKB1 immunohistochemical expression, the control lung tissue showed uniform LKB1 expression. LKB1 protein was present in all the epithelial cells lining the trachea and bronchus, except in goblet cells, whereas it was almost undetectable in the alveolar pneumocytes. LKB1 was highly expressed in the apical surface (i.e., cilia), which is consistent with LKB1’s known roles in the establishment and maintenance of epithelial polarity.

Moreover, 4 μm thick serial sections of each FFPE block of tumor were cut, mounted on non-coated glass slides, and routinely stained with hematoxylin and eosin to perform tissue microdissection. An adequate number of tumor cells from each mounted tumor section was collected using a piezoelectric microdissector (MicroDissector; Eppendorf AG, Hamburg, Germany) and subsequently deposited in a pellet. mRNA transcript levels of LKB1 and NEDD9 were evaluated in all 248 specimens by real-time quantitative reverse transcription polymerase chain reaction. Mutations analysis, DNA extraction, KRAS (codons 12 and 13) and BRAFV600E mutation analysis (by Sanger sequencing) [Appendix A] were performed as previously reported [52,53,54].

#### 2.2.1. IHC Evaluation of LKB1 Expression Status

It is necessary to comment on the strategy implemented for the IHC evaluation of LKB1 expression status. To overcome the fact that immunohistochemical staining of total LKB1 protein does not distinguish between functional and non-functional protein, we employed the use of phosphorylated AMPK at thr172 (pAMPKthr172) to validate our interpretation of the status of expression of LKB1, as it has been shown [5] that LKB1 and pAMPK (Thr172) are interrelated biomarkers that are useful for interrogating LKB1 status of expression in human lung tumors [55]. In fact, the overwhelming majority of the tumors presented almost overlapping areas of LKB1 and pAMPK immunohistochemical staining, with similar degrees of heterogeneous staining intensity ranging from none to strongly cytoplasmic (0–3). This heterogeneity could be interpreted biologically as a reflection of the stages of the evolutionary pattern of LKB1 loss as a tumor suppressor alongside tumor progression. The juxtaposition of the variable combinations of heterogenous staining intensities of LKB1 and pAMPK in the different areas of the tumor in each of the 248 tumor samples yielded a statistically significant association between the status of expression of the two markers, Kendall’s tau 0.524, *p* < 0.001, convalidating our interpretation of LKB1 status expression Figure 1. Even so, it is noteworthy that using IHC to test for the loss of expression of a protein by IHC is definitely more challenging than confirming its presence. Interpreting LKB1 status of expression based on IHC is demanding, with pitfalls to overcome, such as the heterogeneous staining patterns, excessive background staining, and the ambiguous staining of internal positive controls. To date, notwithstanding several reports of LKB1 immunodetection in defined research settings, a widely adopted scoring methodology with optimal accuracy has yet to be selected.

#### 2.2.2. Staging of Lymph Node Metastasis

Pathological lymph node staging was implemented according to the 8th edition of the AJCC [40]. Accordingly, regional (LN1) metastatic involvement is referred to as the metastatic infiltration of ipsilateral peribronchial and/or ipsilateral hilar lymph nodes as well as of intrapulmonary nodes, including involvement by direct extension. Distant (LN2) lymph node involvement refers to metastatic infiltration of ipsilateral mediastinal and/or subcarinal lymph node(s), and distant (LN3) involvement refers to the presence of metastasis in contralateral mediastinal, contralateral, hilar, ipsilateral or contralateral scalene, or supraclavicular lymph node(s) [39].

#### 2.2.3. Statistical Considerations

This is a retrospective translational research study evaluating the expression of different proteins and genes in tumor cells from patients with resectable NSCLC. Due to the observational nature of the study, it was not possible to provide a specific hypothesis to define a sample size estimation. Statistical analysis was executed using R software version 4.3.0 (R core team, Vienna, Austria). Chi-square or Fisher’s exact test was used to compare qualitative variables. Overall survival (OS) was calculated from the date of diagnosis of disease to the date of death from any cause. Median OS (mOS) was estimated using the Kaplan–Meier method (a two-sided log-rank test). A univariate Cox regression analysis, with hazard ratios and 95% CIs, was used to assess the association between each potential predictive factor and survival. These factors were then included in a multivariate Cox proportional hazards regression model with a stepwise procedure to evaluate the independent significance of different variables on survival. All results were considered statistically significant if *p* < 0.05 (two-sided test) level.

## 3. Results

### 3.1. Patients’ Characteristics and Clinicopathological Features

The main demographic and clinical characteristics of the population of our study are summarized in Appendix A. The 248 patients were predominately males 89% (*n* = 220), <70 years of age, 73% (*n* = 181), 76% (*n* = 188) of which presented lymph node metastatic involvement. A total of 48% (*n* = 120) of the primary lung tumors were diagnosed as LUACs, 10 of which (4%) were pleomorphic LUACs; 49.2% (*n* = 122) were diagnosed as Lung Squamous cell carcinoma (LSCC), 3 of which were classified as pleomorphic LSCC (1.2%); 3 (1.2%) were diagnosed as Adenosquamous carcinomas; *n* = 2 (0.8%) as pleomorphic (spindle cell and large cell, one of each) carcinomas; and *n* = 1 case (0.4%) was diagnosed as large cell carcinoma. More than half, 61% (*n* = 151), of the tumors were poorly differentiated grade III tumors and 49% (*n* = 121) were of IIIA p-stage.

At the time of analysis and after a median follow-up of 25.7 months (min–max: 1.3–225.9 months); 156 (63%) disease relapses and 193 (77%) deaths were recorded.

### 3.2. Laboratory Analysis and Correlations

#### 3.2.1. Overall Results of the Laboratory Analysis

The results of the laboratory analysis are presented in Appendix A. Immunohistochemical evaluation of LKB1, pAMPK, PD-L1, STING, p16, p53, CD24, Cyclin-D1, ZEB1, VEGFC, PDGFRα, PDGFRβ, and β-catenin protein expression was successfully performed in all 248 tumor specimens, as shown in Figure 2 and Figure 3. Sequencing for LKB1 and NEDD9 RNA levels was also successfully implemented in all 248 specimens, whereas KRAS exon 2 and BRAF exon 15 mutation analysis was performed in 248 (100%) and 244 (98.3%) specimens, respectively.

LKB1 loss was observed in 21% of specimens (51/248); p16 downregulation was detected in 71% (177/248); low or no PD-L1 expression was detected in 64% (159/248); Cyclin D1 overexpression was detected in 73% (182/248). High expression of PDGFRβ and PDGFRα in the tumor stroma was observed in 83% (205/248) and 74% (183/248), respectively. Ultimately, ZEB1 was highly expressed in 64% (158/248) of the tumors overall. Almost equal percentages of loss and intact expression were observed for STING, p53, VEGFC, CD24, PDGFRα, and PDGFRβ in tumor cells, for ZEB1 in the tumor stroma, as well as for LKB1 and NEDD9 RNA levels. KRAS mutations were detected in 15% (37/248) of the tumors and consisted of G12V (*n* = 9), G13D (*n* = 7), G23D (*n* = 6) G12S (*n* = 5), G12D (*n* = 5), G12C (*n* = 5), G12A (*n* = 2), G13R (*n* = 1), and double G12D + G23D (*n* = 1). BRAF mutations were harbored by 8.2% (20/248) of the tumors, consisting preponderantly of V600E (x17) and to a lesser extent in V600E + K601E (*n* = 2) and V600K (*n* = 1), as shown in Appendix A.

#### 3.2.2. Pleomorphic vs. Non-Pleomorphic LUACs

Confronting non-pleomorphic with pleomorphic LUACs, we assessed statistically significant correlations related to the pleomorphic—dedifferentiated—histology [Appendix A]. Dedifferentiation in LUACs correlated with high PD-L1 expression (*p* < 0.001), with no/low β-catenin membranous staining (*p* = 0.024), with high ZEB1 expression in the tumor (*p* = 0.017), and with KC status (*p* = 0.034). Even though LKB1 loss characterized half,(50% (5/10)), of the pleomorphic LUACs in our study, vs. only one-third (33% (36/110)) of the non-pleomorphic LUACs yielding LKB1 loss, no statistical significance (*p* = 0.3) was established between dedifferentiation and LKB1 loss of expression. Similarly, although 40% (4/10) of the pleomorphic LUACs harbored KRAS mutations versus the 21% (23/110) KRAS mutations present in the non-pleomorphic group, again, no statistical significance was recognized. BRAF mutations were present in almost equal percentages in both pleomorphic and non-LUAC cohorts, 10% (10/110) in the non-pleomorphic LUACs vs. 9.8%, (1/10). Interestingly one-third, (30% (3/10)), of the pleomorphic LUACs harbored KL mutational status.

#### 3.2.3. Pleomorphic vs. Non-Pleomorphic LSCCs

The only laboratory variable correlating with statistical significance with dedifferentiation in pleomorphic LSCCs was the high expression of PD-L1. Specifically, 100% (3/3) of the pleomorphic LSCCs yielded high expression of PD-L1 versus only 39% (47/119) of tumors in the non-pleomorphic cohort (*p* = 0.05) [Appendix A], as shown in Figure 4. Loss of LKB1 expression did not correlate with dedifferentiated LSCCs, as none (0%) of the three (3/3) pleomorphic LSCCs in our group presented LKB1 loss compared to 5.9% (7/119) non-pleomorphic LSCCs with loss of LKB1 expression (*p* > 0.9). Notably, in contrast to the pleomorphic LUACs, no statistically significant correlation was shown between pleomorphic LSCCs and no/low β-catenin membranous staining.

### 3.3. Characterization of NSCLC Tumors with the LKB1-Less Phenotype 

#### 3.3.1. Correlation of LKB1 Loss of Expression with Patients Clinical Characteristics and Tumor Histotype

Loss of LKB1 expression correlated significantly with age, as 82% (42/51), of the tumors with LKB1 loss were obtained from patients under the age of 70 (*p* = 0.091), and of male gender, 78% (40/51), (*p* = 0.009) [Appendix A]. Furthermore, almost half (47% (24/51)) of the tumors with LKB1 loss were p-stage IIIA, while 24% (12/51) were p-stage IIB, and only 16% (8/51) were p-stage IIA. More than half (67% (34/51)) were poorly differentiated grade 3 tumors, and 66% (27/120) of the LUACs (both non-pleomorphic and pleomorphic) had a solid predominant histological pattern. Even though the clear majority (80% (41/51)) of the tumors characterized by loss of LKB1 expression presented concomitant lymph node metastasis LKB1 loss did not show a statistically significant correlation with lymph node metastatic status. Metastatic lymph node dissemination involved mostly regional, LN1, lymph node stations (this was true of 47% (24/51)), but also implicated the coexistence of both regional and distant (LN1 and LN2) lymph nodes, while only 2% (1/51) involved LN3 lymph node stations [Appendix A]. 

As already mentioned above, loss of LKB1 expression was observed in all distinctive tumor histotypes, although as expected, the overwhelming majority of the tumors exhibiting loss of LKB1 expression were LUACs, of which 71% (36/51) were non-pleomorphic and 9.8% (5/51) pleomorphic. LKB1 loss was also observed in 14% (7/51) of the LSCCs, all of which were non-pleomorphic LSCCs. Furthermore, loss of LKB1 expression characterized also 33% (1/3) of the adenosquamous carcinomas, and in addition, the pleomorphic large cell carcinoma (100% (1/1)), the pleomorphic spindle cell carcinoma (100% (1/1)), and the large cell carcinoma included in our study (100% (1/1)) (*p* < 0.001).

Interestingly, high percentages of LKB1 loss of expression was observed and characterized the less-differentiated tumor histotypes. Specifically, half (50% (3/6)) of the pleomorphic LUACs yielded loss of LKB1 expression (vs. 32.7% (36/117) LKB1 loss in the non-pleomorphic LUACs). Likewise, the total (100% (1/1)) pleomorphic large cell carcinoma, (1/1) large cell carcinoma, and also 1 of the 3 adenosquamous carcinomas showed loss of LKB1 expression. Instead, none of the three pleomorphic LSCCs yielded LKB1 loss (0% (0/3)).

#### 3.3.2. Overall Number of Tumors with LKB1 Loss: Correlations with the Examined Laboratory Variables

Overall, loss of LKB1 expression was in absolute agreement with pAMPK loss of expression 100% (51/51) (*p* < 0.001) and yielded a statistically significant correlation with STING loss, since 71% (36/51) of the overall tumors exhibiting LKB1 loss presented concomitant loss of STING expression vs. 44% (87/197) STING loss in the LKB1-intact tumors (*p* < 0.001) [Appendix A], Figure 5.

As expected, LKB1 loss correlated with KRAS mutations, which were harbored by one-third (31% (16/51)) of the tumors with loss of LKB1 expression vs. 11% (21/197) of KRAS mutations present in the LKB1-intact tumors (*p* < 0.001). Significant correlation was also observed, with low LKB1 RNA levels as the clear majority: 69% (35/51) of the tumors that were evaluated by immunohistochemistry to exhibit loss of LKB1 expression were found to also entertain low levels of LKB1 RNA, 69% (35/51), versus the 45% (89/197) LKB1-intact tumors exhibiting low RNA levels in (*p* = 0.003). Moreover, LKB1 loss correlated significantly with high CD24 expression, with 71% (36/51) of the LKB1 loss tumors yielding high CD24 expression vs. the 45% (99/197) that yielded high CD24 expression in LKB1-intact tumors (*p* = 0.009). Similarly, LKB1 loss correlated significantly with the KC co-mutational status, since 24% (12/51) of KC tumors exhibited LKB1 loss versus the 9% (4/6) of KCs that yielded intact LKB1 expression; (*p* < 0.001). BRAF mutations were observed in 14% (7/51) of the tumors with LKB1 loss vs. 6.5% (13/197) in tumors with LKB1-intact expression (*p* = 0.15). No statistically significant correlations were observed among the KP cohort, as only one of the fourteen tumors co-occurred with LKB1 loss (*p* = 0.3). Interestingly, the single tumor in our study with KPL triple co-mutation presented LKB1 loss by definition, so this did not have statistical significance. Additionally, even though the majority of the tumors showing LKB1 loss were characterized by no/low PD-L1 expression —73% (37/51) vs. 62% (122/197) in LKB1-intact tumors—this did not have statistical significance (*p* = 0.2). Ultimately, no statistically significant correlation was observed between LKB1 status and the β-catenin status of membranous expression (*p* > 0.9).

#### 3.3.3. LUACs with LKB1 Loss: Correlations with the Examined Laboratory Variables

Likewise, in LUACs, LKB1 loss correlated with statistical significance with pAMPK (*p* < 0.001) and STING (*p* < 0.001) loss, as well as with KC co-mutational status (*p* = 0.004). Interestingly, contrary to the overall findings in LUACs, LKB1 loss correlated significantly with no/low β-catenin membranous expression (*p* = 0.019), as shown in Figure 6. Meanwhile, the overall correlation observed between LKB1 loss and CD24 was not shown in LUACs, as almost equal percentages of high CD24 expression were observed in LUACs with intact (76%) and tumors showing loss of LKB1 expression (71%) (*p* = 0.5). KRAS mutations were harbored by 37% (15/41) of LUACs showing LKB1 loss vs. 15% (12/79) in the LKB1-intact cohort (*p* = 0.008). BRAF mutations were detected in 15% (6/41) of LUACs with LKB1 loss vs. 6.4% (5/79) in the LUACs with intact LKB1 expression (*p* = 0.2). LKB1 loss correlated significantly with low LKB1 RNA levels (*p* = 0.047) and PD-L1 expression in LUACs did not correlate with the status of LKB1 expression (*p* = 0.3) [Appendix A].

#### 3.3.4. LUACs with LKB1 Loss—Correlations with Clinicopathological Characteristics

Of the 41 total LUACs with LKB1, more than half (66% (27/41)) were characterized by a predominantly solid histological pattern, 20% (8/41) by acinar, and 9.8% (4/41) presented with a predominantly papillary pattern Meanwhile, 12% (5/41) of the total LUACs with LKB1 loss were pleomorphic LUACs. More than three-quarters were accompanied by lymph node metastases, which similar to overall showed a lack of correlation with the status of LKB1 expression [Appendix A].

#### 3.3.5. LSCCs with LKB1 Loss—Correlations with the Examined Laboratory Variables

Of the 122 LSCCs involved in our study, only 5.77% (7/122) yielded LKB1 loss. The only statistically significant correlation detected in the LSCCs with LKB1 loss was loss of p-AMPK expression, which was observed in 100% the LSCCs (7/7), (*p* < 0.001). Even though the overwhelming majority of LSCCs tumors were characterized by concomitant loss of LKB1 and STING expression (86% (6/7) versus 51% (59/115) STING loss in LKB1-intact LSCCs tumors) this correlation did not gain statical significance (*p* = 0.12), most likely due to the very small number of tumors representative of this category (*n* = 7). Notably, 0% of the seven (0/7) LSCCs showing LKB1 loss harbored KRAS mutations compared to the 7% (8/119) of KRAS mutant tumors in the LSCCs with intact LKB1 expression. Instead, BRAF mutations were detected in 14%, (1/7) of the LSCCs with LKB1 loss versus 7.1% (8/115) in the LSCCs with LKB1 intact expression (*p* = 0.4), the total exclusively harbored by non-pleomorphic LSCCs. PD-L1 expression did not correlate with LKB1 status, as almost equal percentages of LKB1 loss were observed in both LSCCs with loss and IN LSCCs with intact LKB1 expression. Likewise, PD-L1 expression was similar in pleomorphic and non-pleomorphic LSCCs.

#### 3.3.6. LSCCs with LKB1 Loss—Correlation with Clinicopathological Characteristics

A total of 86% (6/7) of the LSCCs with LKB1 loss presented concomitant lymph node metastasis. Most of these (71% (5/7)) involved regional lymph nodes, whereas 14% (1/7) involved both regional and distant (LN1 and LN2) lymph node stations. Interestingly, the majority (71% (5/7)) were p-stage I–II, although poorly differentiated grade 3 tumors, and most of the patients (71% (5/7) were <70 years of age. None of the LSCCs with LKB1 loss were pleomorphic.

### 3.4. Correlation of LKB1 Loss and Lymph Node Metastatic Status 

#### 3.4.1. Metastatic NSCLCs with LKB1 Loss vs. Metastatic NSCLC with LKB1 Intact in Relation to the Examined Laboratory Variables

Overall, 41/188 patients with lymph node metastases yielded tumors with concomitant loss of LKB1 expression. Importantly, in metastatic tumors, loss of LKB1 protein expression was found to correlate significantly with concurrent loss of STING, (*p* = 0.014) and p-AMPK expression; with KRAS mutations, 39% (13/147 vs. 19/147); with KC status, 29% (12/41), (*p* < 0.001); with low levels of LKB1 RNA (*p* = 0.008); and with high CD24 expression (*p* = 0.025). No statistically significant correlations were observed between metastatic tumors with LKB1 loss and β-catenin membranous expression status, or with PD-L1 expression. BRAF mutations were harbored by (12%) 5/41 vs. (7.7%) 11/147 of the metastatic tumors with intact LKB1 expression. The majority (71% (29/41)) of the 41 metastatic tumors showing LKB1 loss were non-pleomorphic LUACs, while 9.8% (4/41) were pleomorphic LUACs, and 15% (6/41) were LSCCs [Appendix A].

#### 3.4.2. Metastatic LUACs with LKB1 Loss vs. Metastatic LUACs with LKB1 Intact in Relation to the Examined Laboratory Variables

More than three-quarters (80.5% (33/41)) of the metastatic tumors with LKB1 loss were LUACs. Similarly, in metastatic LUACs, LKB1 loss correlated significantly with STING loss (*p* = 0.005), with p-AMPK loss (*p* < 0.001), with KRAS mutations 45% (15/33) [vs. 20% (11/56) in the metastatic LUACs with intact LKB1 expression] (*p* = 0.010), and with KC status (*p* = 0.009), while in contrast to overall findings in metastatic LUACs showing LKB1 loss, a significant correlation was observed with no/low β-catenin membranous expression (*p* = 0.007). No statistically significant correlation with PD-L1 expression was observed. BRAF mutations were detected in 22% (4/33) vs. 7.3% (4/56) in the metastatic LUACs with intact LKB1 expression [Appendix A].

#### 3.4.3. Metastatic LSCCs with LKB1 Loss vs. Metastatic LSCCs with LKB1 Intact in Relation to the Examined Laboratory Variables 

Even though of the seven total metastatic LSCCs, six yielded concomitant LKB1 loss, 86% (6/7), the only statistically significant correlation observed was with pAMPK loss (*p* < 0.001). Interestingly, even though STING loss was observed in 83% (5/6) of the metastatic LSCCs with LKB1 loss vs. 57% (51/89) in the metastatic LSCCs with intact LKB1expression, this did not gain statistical significance (*p* = 0.4) most likely attributable to the rarity of LSCCs showing LKB1 loss of expression (*n* = 7) [Appendix A].

### 3.5. KRAS Co-Mutational Cohorts—Incidence

KL co-mutation characterized 6.5% (16/248) of the NSCLC tumors included in our study, KC status 8.5% (21/248), KRAS mutant and LKB1 intact status 8.46% (21/248), KP status 5.6% (14/248), L status 2.4%, K status 1.6% (4/248), and finally, the triple-mutated KPL status only 0.4% of the tumors (1/248) [Appendix A].

#### 3.5.1. Clinicopathologic and Laboratory Outline of KRAS Co-Mutational Cohorts

KL cohort (*n* = 16): The preponderance of the tumors with KL co-mutational status were characterized by p16 downregulation, 75% (12/16) and by a “wild type” p53 immunohistochemical pattern of expression, 94% (15/16). Moreover, 94% (15/16), showed high expression of PDGFRβ, 88% (14/16) exhibited the presence of PDGFRa, and 81% (13/16) showed similar expression in the tumor stroma, while high CD24 expression in the tumor was observed in 63% of cases (10/16). Furthermore, more than half (69% (11/16)) showed no/low β-catenin membranous expression. Similar percentages of STING loss and intact expression were observed: 56% (10/16) and 44% (7/10), respectively. Ultimately, BRAF commutation was detected in 6.3% of the tumors in this cohort (1/16).Overall, 75% (12/16) of the patients with KL tumors were males, and 94% (15/16) were aged <70 years. Interestingly, the KL tumor cohort did not include LSCCs and consisted almost exclusively of non-pleomorphic and pleomorphic LUACs, [75% (12/16) and 19% (3/16), respectively], with solid, 60% (9/16), and acinar, 27% (4/16) as the predominant histological patterns. Though the tumors showed various grades of differentiation, more than half (56% (9/16)) were p-stage III–IV.KC (*n* = 21): KC tumors yielded almost equal percentages of lost and intact LKB1 expression: 57% (12/21) vs. 43% (9/21), respectively, even though, curiously, more than three-quarters (81% (17/81)) of the tumors showed low LKB1 RNA levels. All of them (100% (21/21)) exhibited p16 downregulation. High expression of PDGFRβ and PDFGRα in the tumor stroma characterized 90% (19/21) and 86% (18/21) of the tumors, respectively. High CD24 expression was observed in 71% (15/21) and Cyclin D1 overexpression was observed in 86% (18/21). No/low β-catenin membranous expression was shown in 57% (8/21). Interestingly, 10% of the tumors with KC co-mutation (2/21) presented concomitant BRAF mutations. More than half of the KC status tumors (62% (13/21)) were p-stage IIIA, grade 3 poorly differentiated tumors, with 57% (12/21) exhibiting a size of >5 cm. Meanwhile, 57% (14/21) of KC tumors were LUACs, 62% (13/21) were non-pleomorphic, and 19% (4/21) were pleomorphic, including an invasive mucinous histological pattern (5.9%). The cohort also included LSCCs (14% (3/21)) and one (1/21) adenosquamous carcinoma (*p* > 0.002).

The laboratory and clinical characteristics of KRAS mutant and LKB1 intact, KP, L, KPL are described in detail in Appendix A.

#### 3.5.2. KRAS Co-Mutational Cohorts—Correlation with Lymph Node Metastatic Status

Notably, all mono-, bi-, and tri-mutational cohorts yielded significant percentages of lymph node metastasis, 90% (75/83), with statistically significant correlations between the mutational cohorts (*p* = 0.032).

KL and KC bi-mutational status correlated significantly with lymph node metastasis, as all (16/16) of the KL tumors and all but one (20/21) KC tumors were associated with lymph node metastasis 8.5% (16/188) and 10% (20/188) vs. 0% (0/60) and 1.7% (1/188) (*p* = 0.015) and (*p* = 0.03), respectively. On the contrary, KP status did not show a correlation with lymph node metastasis (*p* = 0.2). Specifically, KL and KPL cohorts yielded the highest percentage (100%) of concomitant lymph node metastasis (16/16 and 1/1, respectively) with none of the tumors in any cohort having intact lymph nodes, LN0. Additionally, 95% (20/21) of the KC, 93% (13/14) of the KP, and 90% (19/21) of the KRAS mutant and LKB1-intact tumors presented lymph node metastasis. The lowest percentages of lymph node metastasis were observed in the mono-mutational cohorts L and K with 75% (3/4) and 50% (3/6), respectively. In particular, 100% of the KPL (1/1) cohort, almost half of the KL and KC cohorts (44% (7/16) and 43% (9/21), respectively), one-third of the KP and KRAS mutant LKB1-intact cohorts (29% (4/14) and (6/21), respectively), and one-quarter of the K cohort (25% (1/4)) yielded metastasis in both regional and distant mediastinal (LN1 and LN2) lymph nodes. On the contrary, only half (50% (3/6)) of the tumors in the L cohort presented metastatic involvement, which was associated with exclusively regional LN1 lymph nodes. Interestingly, the only mutational subgroup presenting with LN3 involvement (supraclavicular lymph node in our case) was KL. KL yielded superior aggressiveness, with no tumors of LN0 status, exhibiting LN3 metastatic involvement besides the highest percentage of combined LN1+ LN2 involvement 44% (7/16).

### 3.6. Survival Analysis

#### 3.6.1. Overall Survival Analysis—mOS

The univariate analysis in the overall cohort revealed that patients with tumors yielding loss of STING expression experienced significantly shorter median overall survival (mOS) compared to patients with tumors showing intact STING expression: 19.68 vs. 36.53 months; *p* = 0.004, as shown in Figure 7 [Appendix A].

Likewise, patients with tumors showing p53 mutational immunohistochemical expression status had significantly inferior mOS compared to patients with p53 intact expression: 19.83 vs. 34.76 months; *p* = 0.029. Similarly, patients with tumors harboring KRAS mutations had inferior mOS compared to KRAS wild-type tumors: 19.88 vs. 28.25 months; *p* = 0.037. Also, tumors with high PDGFRβ expression were associated with a distinct decrease in mOS compared to tumors with low PDGFRβ expression, with survival rates of 20.42 vs. 36.34 months; *p* < 0.001. Additionally, patients with tumors yielding no/low β-catenin membranous expression were associated with a clear decrease in mOS compared to tumors with intermediate/high β-catenin membranous expression: 20.70 vs. 36.40 months; *p* = 0.003. Furthermore, a clear decrease in mOS was observed in patients with tumors exhibiting high ZEB1 expression compared to that of patients with tumors with low ZEB1 expression: 21.45 vs. 37.29 months; *p* = 0.045. Ultimately, as expected, patients with metastatic involvement of regional and/or distal lymph nodes experienced a significant decrease in mOS compared to patients with intact lymph nodes: 21.29 vs. 46.75 months; *p* < 0.001. A trend of decreased mOS was observed in relation to Cyclin D1 overexpression and normal expression, with mOS of 20.09 vs. 33.23; *p* = 0.056, while high PDGFRβ expression in the tumor stroma resulted in survival rates of 24.90 vs. 48.36 months; *p* = 0.068. LKB1 loss vs. intact status did not show a significant correlation with mOS (*p* > 0.9)

#### 3.6.2. Overall Survival Analysis—Hazard Ratio Univariate Analysis

Univariate analysis revealed that patients with tumors showing intact STING expression experienced a lower risk of death (HR: 0.66, 95% c.i.: 0.50–0.88; *p* = 0.005). On the contrary, patients with tumors yielding high PDGFRb expression (HR: 1.71, 95% c.i.: 1.28–2.28; *p* < 0.001), patients with metastasis to regional and distant lymph nodes (HR: 1.83, 95% c.i.: 1.27–2.64; *p* = 0.001), patients with tumors showing no/low β-catenin membranous expression (HR: 1.53, 95% c.i.: 1.15–2.03; *p* = 0.004), patients with KRAS mutations (HR: 1.50, 95% c.i.: 1.02–2.21; *p* = 0.039), patients with high ZEB1 expression in the tumor (HR: 1.36, 95% c.i.: 1.01–1.84; *p* = 0.046), with Cyclin D1 overexpression (HR: 0.74, 95% c.i.: 0.54–1.01; *p* = 0.057), as well as patients with tumors exhibiting p53 mutational immunohistochemical expression status (HR: 1.37, 95% c.i.: 1.03–1.83; *p* = 0.030) were at significantly higher risk of death.

#### 3.6.3. Overall Survival Analysis—Multivariate Analysis

Furthermore, in multivariate analysis, STING intact expression emerged as an independent prognostic factor for increased survival with a significantly decreased risk of death (HR: 0.61, 95% c.i.: 0.45–0.82; *p* < 0.001). Instead, high PDGFRβ expression in the tumor and KRAS mutations emerged as independent prognostic factors for increased risk of death: (HR:1.62, c.i. 1.16–2.24; *p* = 0.004) and (HR: 1.81, 95% c.i.: 1.20–2.72; *p* = 0.005), respectively.

Loss of β-catenin membranous expression and p53 mutational expression status yielded a trend of correlation to a higher risk of death; (HR: 1.29, c.i. 0.96–1.74; *p* = 0.094) and (HR: 1.29, c.i. 0.96–1.74; *p* = 0.091). Metastasis to regional and/or mediastinal lymph nodes as well as high ZEB1 expression in the tumor did not reveal as independent prognostic for higher risk of death (HR: 1.15, c.i. 0.73–1.82; *p* = 0.539) and (HR: 1.06, c.i. 0.76–1.48; *p* = 0.539), respectively—Figure 8.

#### 3.6.4. Tumors with LKB1 Loss of Expression—Median Overall Surviival

The univariate analysis in the cohort of tumors with LKB1 loss revealed that patients with tumors yielding concomitant LKB1 loss and no/low β-catenin membranous expression experienced significantly decreased median survival mOS compared to tumors showing LKB1 loss and intact β-catenin membranous expression: 20.50 vs. 52.99 months; *p* < 0.001. All other comparisons did not reveal any significant correlations between the expression of the various proteins, RNA protein levels, or the presence of KRAS, and BRAF mutations with mOS (all long-rank *p* > 0.05), Figure 9. Appendix A.

#### 3.6.5. Tumors with LKB1 loss —Univariate Analysis Hazard Ratio

Importantly, the hazard ratio analysis revealed that patients with concurrent LKB1 loss and no/low β-catenin membranous expression experienced a significantly increased risk of death (HR: 3.32, 95% c.i.: 1.71–6.43; *p* =0.00002) Figure 10.

#### 3.6.6. KL Tumors Cohort Survival Analysis—mOS Univariate Analysis

KL tumors—univariate analysis of the KL cohort revealed that patients with KL tumors yielding concomitant loss of STING expression experienced significantly shorter median overall survival (mOS) compared to patients with KL status tumors showing intact STING expression, 18.73 vs. 52.99 months; *p* = 0.011 Figure 11.

Likewise, patients harboring the KL status and concomitant no/low β-catenin membranous expression showed a trend correlating with decreased mOS vs. patients of KL status with retained, medium–high β-catenin membranous expression: 20.07 vs. 52.99; *p* = 0.069, Appendix A.

#### 3.6.7. KL Tumors—Univariate Analysis, Hazard Ratio

Furthermore, patients with KL mutational status and concurrent intact STING expression experienced a significant decrease in the risk of death (HR: 0.12, 95% ci: 0.05–0.76; *p* = 0.02). Inversely, patients with KL tumors yielding no/low β-catenin membranous expression showed a trend of increased risk of death (HR: 3.18, 95% ci: 0.86–110.78; *p* = 0.08), as shown in Figure 12.

## 4. Discussion

In this study, we performed an in situ immunohistochemical investigation of the status of LKB1 expression in all NSCLC histotypes, beyond and within *KRAS* mutational status, so as to investigate the LKB1-less phenotype and its possible correlations, as well as the impact of LKB1 status, the emergence of significant correlations, and tumor patients’ prognosis. Furthermore, we aimed to explore the prevalence and the clinical and laboratory characteristics of defined KRAS co-mutational groups, as well as their prognostic impact in patients with operable non-small cell lung cancer (NSCLC).

Loss of LKB1 protein expression was observed in 21% of the overall examined tumors and correlated significantly with histotype (*p* < 0.001). Specifically, loss of LKB1 expression was observed with variable incidences in all the represented distinctive histotypes in our study, except the three pleomorphic LSCCs and the spindle cell pleomorphic carcinoma, which showed intact LKB1 expression.

Overall, tumors exhibiting loss of LKB1 expression were poorly differentiated grade 3 tumors, and were mainly observed in patients of male gender under 70 years of age. Metastatic lymph node dissemination involved mostly regional (LN1) lymph node stations, 47% (24/51), but also implicated both regional and distant (LN1 and LN2) lymph nodes.

Overall, LKB1 loss showed a significant correlation with p-AMPK loss, with KRAS mutations, with low LKB1 RNA levels, with high expression of CD24, with Cyclin-D1 overexpression, and with the KC mutational status, both overall and in LUACs.

The statistically significant correlation between LKB1 loss and the loss of STING expression documented overall (*p* < 0.001), and in LUACs (*p* < 0.001) was also observed in the metastatic tumors, with overall LKB1 loss of (*p* = 0.014), while that in LUACs was (*p* = 0.005).

No statistically significant correlation was established with LSCCs, most likely due to the small number of LSCCs in the LKB1 loss group due to the rarity of tumors representative of this category in our study (*n* = 7).

LKB1 loss showed a significant correlation with β-catenin no/low membranous expression, exclusively in LUACs, overall (*p* = 0.019) and, importantly, in the metastatic setting (*p* = 0.005). Furthermore, no/low β-catenin membranous staining was shown to characterize dedifferentiated—pleomorphic—LUACs (*p* = 0.024).

No significant correlation was observed between LKB1 loss and PD-L1 expression status, overall, in LUACs or LSCCs.

Remarkably, moderate–high PD-L1 expression (scores 2, 3) was observed to correlate with dedifferentiation, as high percentages of PD-L1 expression characterized the overwhelming majority of pleomorphic LUACs and LSCCs, as well as the histologically dedifferentiated pleomorphic tumors (spindle and large cell carcinomas) along with no/low β-catenin membranous expression. The high levels of PD-L1 expression could be related to the significant amount of inflammatory infiltrate that was prevalent in the pleomorphic lung carcinomas.

In contrast, high percentages (ε 50%) of no/low PD-L1 expression characterized all members of the KRAS co-mutational cohorts, with tumors harboring the triple-mutated KPL status showing the highest percentage (100% (1/1)), followed by tumors with KL status (81% (13/16)), KP and KC status tumors (71% (10/14 and 15/21, respectively)), and, ultimately, KRAS mut and LKB1-intact tumors, 67% (14/21).

Interestingly, the obligatory mono-mutational cohorts K and L showed the lowest percentages of no/low PD-L1 expression, at 50% (2/4 and 3/6, respectively), which was hypothesized to be due to the lower mutational load.

Additionally, even though LKB1 loss, per se, did not correlate with lymph node metastasis, when it co-occurred with KRAS mutations (KL) or as a component of KPL status, 100% of the KL and KPL (16/16 and 1/1) tumors showed lymph node metastasis versus only half (50% (3/6)) of the mono-mutational L cohort (LKB1 loss only, KRAS: wild type, p53: wild type, p16: not downregulated) with lymph node metastasis (*p* = 0.032).

Even though the loss of LKB1 expression, did not necessarily affect survival, patients with concomitant LKB1 loss and no/low β-catenin membranous expression experienced significantly inferior median overall survival of 20.50 vs. 52.99 months; *p* < 0.001, as well as significantly greater risk of death (HR: 3.32, 95% c.i.: 1.71–6.43; *p* < 0.001). This was observed in patients with tumors yielding LKB1 loss and intact, intermediate–high β-catenin membranous expression.

Notably, LKB1 loss in co-occurrence with KRAS mutation was the only KRAS co-mutation to implicate the LN3 lymph node station. Conversely, only half (50% (3/6)) of the tumors belonging to the mono-mutational (L) cohort presented lymph node metastasis. The survival analysis of the KL cohort revealed that patients with tumors yielding concomitant loss of STING expression experienced significantly shorter median overall survival (mOS) compared to patients with tumors showing intact STING expression (18.73 vs. 52.99 months; *p* = 0.011), while patients with tumors harboring the KL status and concomitant no/low β-catenin membranous expression showed a trend correlating with decreased mOS vs. patients of KL status with retained intermediate–high β-catenin membranous expression (20.07 vs. 52.99; *p* = 0.069). Ultimately, patients with tumors harboring KL mutational status and concurrent intact STING expression experienced a significant decrease in the risk of death (HR: 0.12, 95% ci: 0.05–0.76; *p* = 0.02), while patients with tumors yielding no/lower membranous expression of β-catenin showed a trend correlated with an increased risk of death (HR: 3.18, 95% ci: 0.86–110.78; *p* = 0.08).

Our results highlight the significant correlation between LKB1 loss and STING loss and are in accordance with the findings of Kitajima et al. [56]. In this seminal study, it was shown that LKB1 loss results in marked silencing of STING, which is mediated—at least in part—by hyperactivation of DNMT1 and EZH2 activity related to elevated S-adenylmethionine levels and reinforced by DNMT1 upregulation. The authors, after identifying KL-specific downregulation of genes involved in dsDNA sensing—especially TMEM173/STING, which was also reduced in LKB1-mutant KRAS wild-type NSCLC cell lines—performed IHC to validate their findings and analyzed tumor cell-specific STING protein levels across a panel of 64 patient-derived NSCLC samples enriched for KRAS-mutated status. The results revealed that LKB1 loss was robustly associated with either a complete absence of or a significant reduction in STING levels in tumor cells. Importantly, the authors found that STING silencing was associated with LKB1 loss also in KRAS WT lung cancers, and was especially robust when combined with elevated DNMT1 levels, suggesting that this mechanism is not purely limited to the KL cellular state. 

Furthermore in our study we show that concurrent LKB1 loss and STING loss of expression correlates with statistical significance with lymph node metastasis in both overall NSCLC tumors but also in the LUACs cohort.

As already described, we observed very high percentages of no/low levels of PD-L1 expression in all KRAS mutational cohorts, with the KL presenting the highest percentages. This finding is also in agreement with [37,56], stating that defective dsDNA sensing links LKB1 loss with low PD-L1 levels. Thus, silencing of STING in KL cells directly contributes to impaired intratumoral T-cell recruitment and low PD-L1 expression.

Furthermore, also in the setting of KRAS-LKB1 co-mutation, we show that patients with KL tumors yielding concomitant loss of STING expression experienced significantly shorter median overall survival (mOS) compared to patients with KL status tumors showing intact STING expression, while patients with KL tumors and concurrent intact STING expression experienced a significant decrease in the risk of death. In multivariate analysis, STING intact expression emerged as an independent prognostic factor for increased survival. Instead, loss of β-catenin membranous expression status yielded a trend correlating with a higher risk of death.

An important finding of our study was the correlation of LKB1 loss with no/low β-catenin membranous expression observed in LUACs, both metastatic and non-metastatic. This correlation yielded significant implications in prognosis, as the survival analysis revealed that patients with concurrent LKB1 loss and no/low β-catenin membranous expression showed significant decreases in mOS and showed a higher risk of death (HR) compared to patients with LKB1 loss tumors and intact β-catenin membranous expression.

Our results are in accordance with data from a recent paper investigating the contribution of four missense LKB1 somatic mutations in the biology of the tumor. The researchers showed that LKB1R87K and, to a lesser extent, LKB1D194Y mutant isoform induced cell motility, promoting cytoskeleton regulation and β-catenin degradation and loss at the cell junctions in the 3D cultures [16].

β-catenin implements its structural function as a member of the catenins—proteins that link the key molecule of Ca^2+^-dependent cell adhesion E-cadherin to cytoskeletal structures [57,58]. Several studies have highlighted the role of LKB1 metastasis, adhesion, and motility [33]. It has been shown that LKB1 serves as a focal adhesion kinase (FAK) repressor, where LKB1 depletion causes rapid focal adhesion site turnover [59] loss in cell polarity and motility through the regulation of PAK115 and the modulation of the phosphorylation status of FAK and CDC42 activation [60] and the contribution of LKB1 loss to the induction of epithelial–mesenchymal transition (EMT) and metastasis [33,34].

Although the results of the current study should be validated in the future, we consider that they are robust and in accordance with the literature. The weaknesses of the study are the retrospective nature of the analysis, the lack of a validation group, and the different management of patients after the surgical resection, since some of them have received adjuvant chemo- and/or radiotherapy while others have not. For this reason, an analysis of the Disease-Free Survival (DFS) was not performed, since this could be misleading. Furthermore, patients’ samples were collected retrospectively even after 10 years, and several therapeutic improvements have been made during this period.

These results require further investigation in prospective studies. Despite this, the presented data provide a better understanding of the deregulations of key pathways downstream of LKB1 and of its substrates, which can expose weak points of LKB1 mutant tumors that may be therapeutically exploited to facilitate our understanding of the natural history of the disease. Immunohistochemical screening on FFPE tissue of both total LKB1 protein and AMPKthr172 as clinical biomarkers may provide an effective alternative approach to costly whole-gene or DNA-based sequencing, as it has the potential to capture most mechanisms conferring protein instability, loss-of-function mutations, and deletions, suggesting that protein degradation can be used as a means to identify the loss of LKB1 in cancer.

## 5. Conclusions

In summary, the results of the present study provide insights into the NSCLC tumors with the LKB1-less phenotype and KRAS co-mutations. Investigating the loss of LKB1 expression may have significant “theranostic” implications in the management of patients with operable NSCLC, and the investigation of LKB1 loss could be a step forward in the procedure of precision medicine in lung cancer. The expression status of LKB1 in association with STING and β-catenin membranous expression status could be used as stratification factors for patients with operable NSCLC.

## Figures and Tables

**Figure 1 cancers-16-01818-f001:**
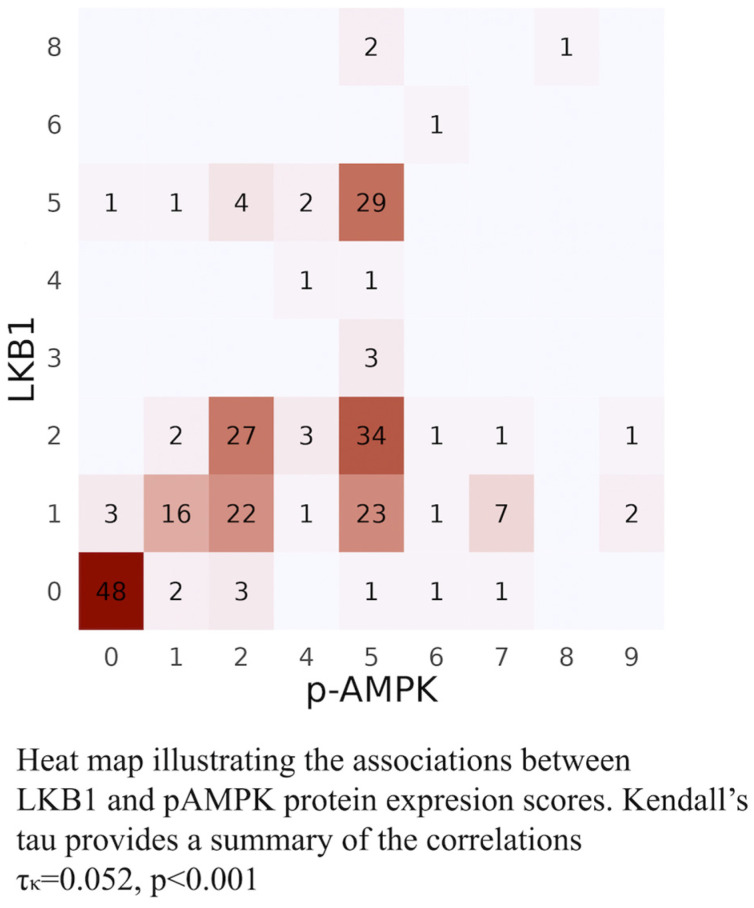
Heat map of LKB1-pAMPK. Combinations of intensities observed in the heterogeneous staining areas of the tumors observed in both LKB1 and pAMPK staining. In total, scores 0–9 were assigned to include as many of the multifaceted combinations of intensities as possible (0 = lost, 1 = low, 2 = moderate, 3 = strong staining present in different areas of the majority of the tumors). [0 = 0, 1 = 0-1, 2 = 0-1-2, 3 = 0-2-3, 4 = 0-2-3, 5 = 0-1-2-3, 6 = 1-2, 7 = 2, 8 = 1-2-3, 9 = 0-3].

**Figure 2 cancers-16-01818-f002:**
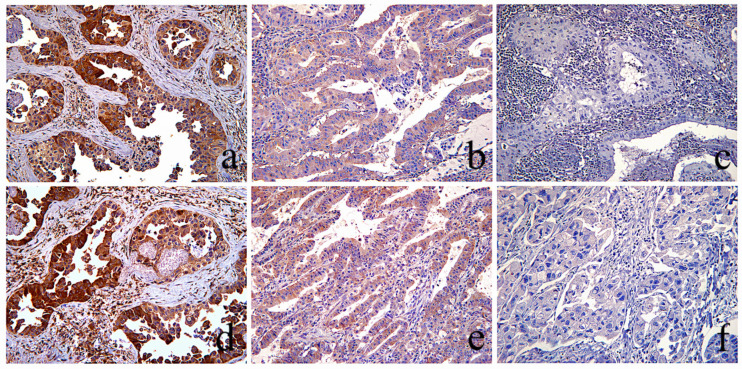
LKB1 and p-AMPK immunohistochemical evaluation. Top row: Three different NSCLC tumors immunohistochemically stained for LKB1. Bottom row: Three different NSCLC tumors immunohistochemically stained against p-AMPK. (**a**,**d**) The same area of a LUAC stained with LKB1 (**a**), and pAMPK (**b**) showing cytoplasmic staining of similar, moderate and strong (scores: 2–3), intensity, therefore evaluated as “intact”. (**b**,**e**) Same area of a LUAC showing similar, very low (score: 1), staining intensity of LKB1 and pAMPK expression, evaluated as “intact”. (**c**,**f**) The same area of a LSCC with complete absence, (score: 0), of staining for both LKB1 and pAMPK was evaluated as “lost”(score: 0); [(**a**–**e**) magnification ×100, (**f**) magnification ×200].

**Figure 3 cancers-16-01818-f003:**
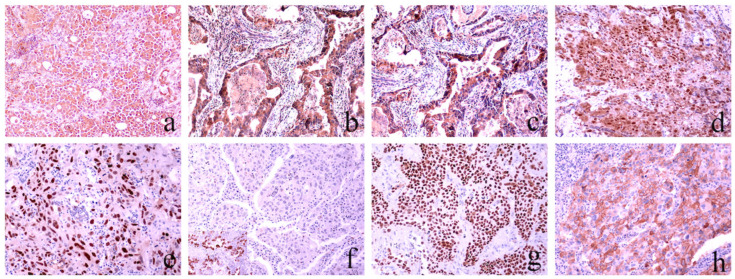
Representative sections of NSCLC tumors immunohistochemically stained for the assessment of VEGFC (**a**), PDGFRα (**b**), PDGFRβ (**c**), ΖΕΒ1 (**d**), Cyclin D1 (**e**), p16 (**f**), p53 (**g**), and CD24 (**h**) expression. For VEGFC, PDGFRα, and PDGFRβ, the cytoplasmic staining was evaluated, while for ZEB1, Cyclin D1, p16, and p53, nuclear staining was assessed. Ultimately, membranous staining was evaluated to determine the expression of CD24. Staining intensity was assessed by comparison using an accepted external/internal positive control [(**a**–**f**): magnification ×200]. (**f**) The f insert shows positive p16 nuclear expression [magnification ×400].

**Figure 4 cancers-16-01818-f004:**
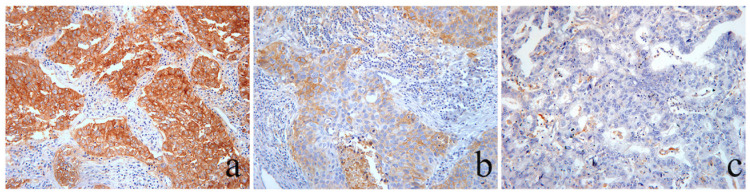
PD-L1 immunohistochemical staining. (**a**) Area of a LUAC of solid predominant histological pattern showing strong membranous PD-L1 staining in 100% of the tumor cells, therefore scored as “high positive”; (**b**) Area from a LUAC showing focally moderate-weak membrane staining in 15% of the tumor cells, scored as “low positive”; (**c**) Area from a LUAC with complete absence of PD-L1 staining 0%, in the presence of internal positive control (immune cells), scored as “negative”. [(**a**,**b**): magnification ×100, (**c**): magnification ×200].

**Figure 5 cancers-16-01818-f005:**
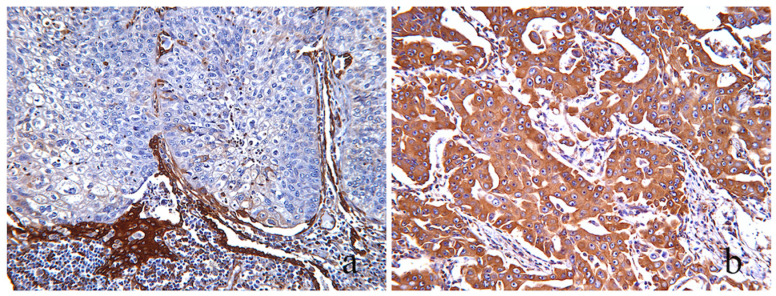
STING immunohistochemical staining. (**a**) An area of a LSCC showing complete loss of STING expression in the presence of positively stained stromal cells and immune cells as internal positive controls. (**b**) An area of a LUAC exhibiting “high”, score: 3, cytoplasmic staining intensity of STING in 100% of the tumor cells. [(**a**,**b**): magnification ×200].

**Figure 6 cancers-16-01818-f006:**
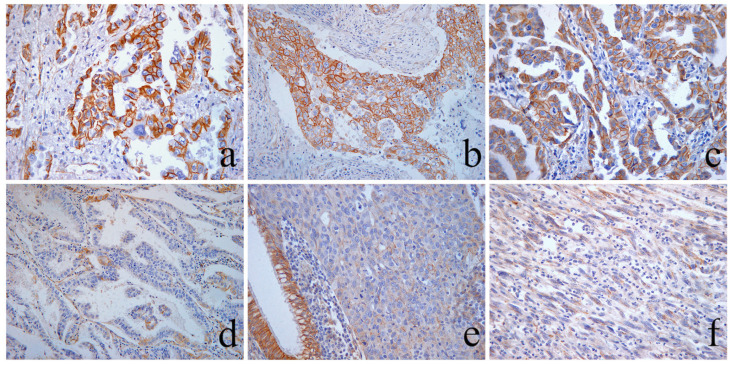
Immunohistochemical evaluation of β-catenin membranous expression. (**a**) Area of a LUAC showing strong complete β-catenin membranous staining in 95% of the tumor cells, scored as 3: “high”. (**b**) Area of an LSCC showing heterogeneous β-catenin membranous expression. There is coexistence of areas of tumor cells that exhibit strong and complete membranous staining (peripheric zones) and areas with tumor cells showing moderate–weak β-catenin membranous expression (centrally). The latter is observed in 60% of the tumor cells, and therefore is scored as 2: “moderate”. Distinct, patchy foci of single cells and small groups of cells with vanishing or lost membranous expression are also present. (**c**) Area of a LUAC with complete membranous staining preservation of moderate–strong intensity in >70% of cells, scored as 3: “high”. (**d**) An area of a LUAC showing vanishing β-catenin membranous expression with only small patchy foci observed in 5% of the tumor cells, retaining a membranous expression score of 1: “low”. (**e**) Area from a grade 3 LUAC with a solid predominant histological pattern showing complete loss of membrane staining of β-catenin, in the presence of the intact membrane staining of the by standing bronchial epithelium as internal positive control. (**f**) Dedifferentiated area morphology of a pleomorphic LUAC with sarcomatous spindle cell exhibiting complete absence of β-catenin membranous expression and consequently scored 0: “lost”. Magnification ×100.

**Figure 7 cancers-16-01818-f007:**
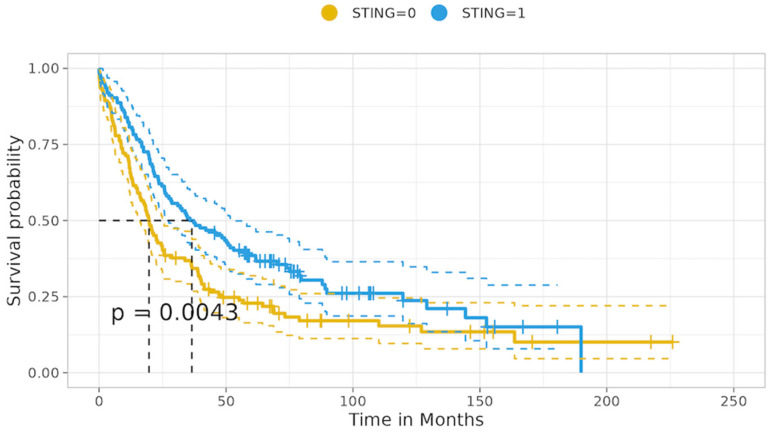
Overall—STING status of expression in relation to median overall survival.

**Figure 8 cancers-16-01818-f008:**
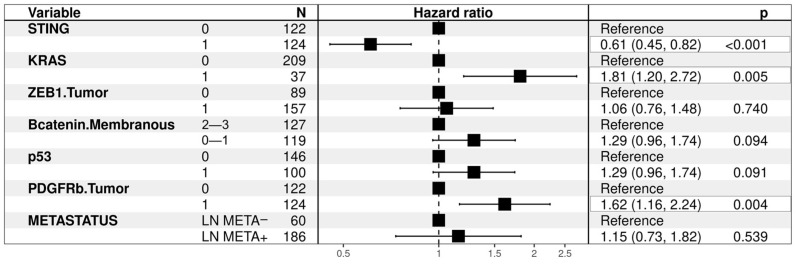
Overall multivariate analysis —Hazard Ratio.

**Figure 9 cancers-16-01818-f009:**
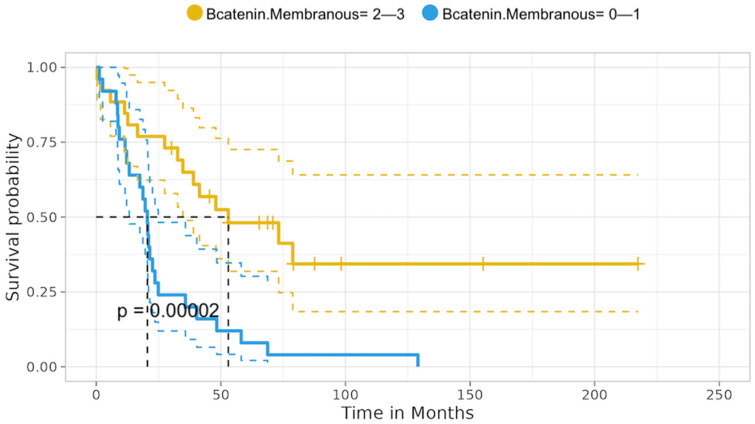
LKB1 loss tumors correlation with β-catenin membranous status of expression—mOS.

**Figure 10 cancers-16-01818-f010:**
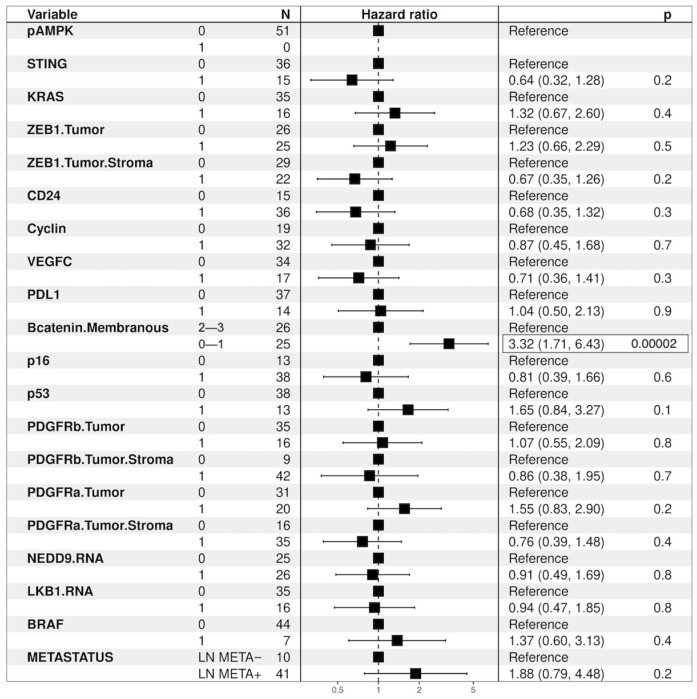
LKB1 loss tumors correlations—Hazard ratio.

**Figure 11 cancers-16-01818-f011:**
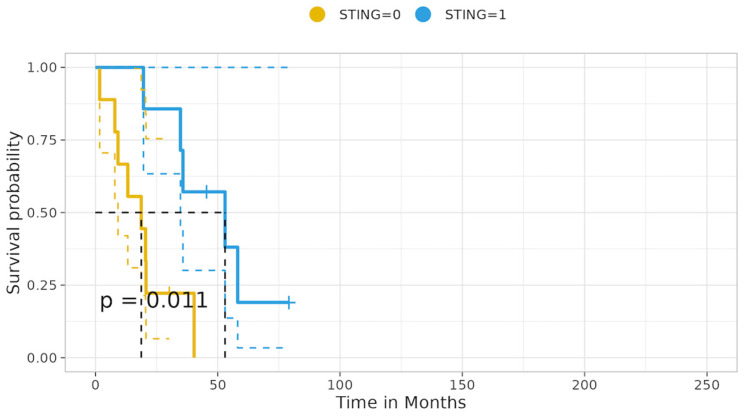
KL status tumors correlation with STING expression status—median overall survival.

**Figure 12 cancers-16-01818-f012:**
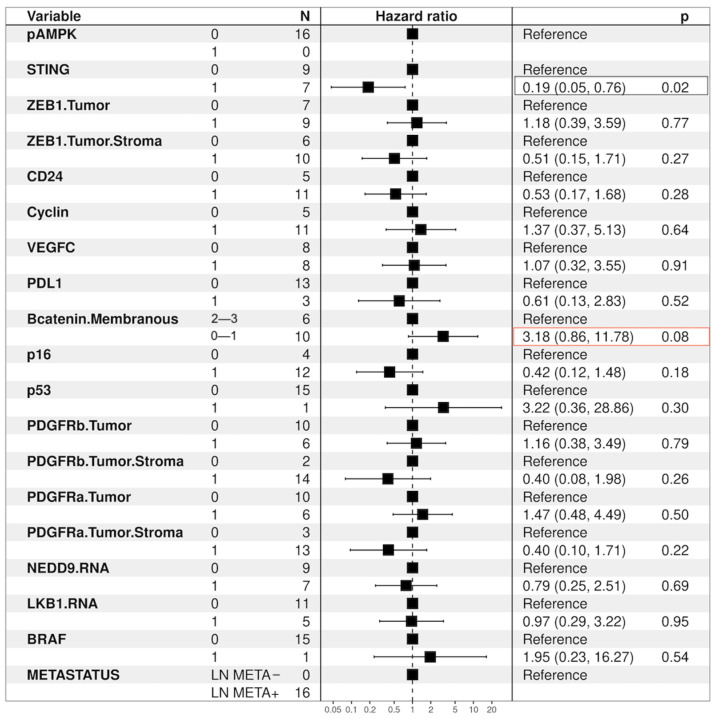
Tumors with the KL co-mutation—Hazard ratio.

## Data Availability

The data that support the findings of this study are available from the corresponding author, E.L., upon reasonable request.

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
