# Peer review of "LKB1 Loss Correlates with STING Loss and, in Cooperation with β-Catenin Membranous Loss, Indicates Poor Prognosis in Patients with Operable Non-Small Cell Lung Cancer"

_cancers, 2024, doi:10.3390/cancers16101818_

Round 1

Reviewer 1 Report

Comments and Suggestions for Authors

Lung cancer is complicated and diverse, with cellular and histological differences. Subgroups and genetic changes complicate diagnosis and treatment. Recent molecular and immunohistochemical advances have improved lung cancer progression-free survival. Research has shown remarkable efficacy in patients with actionable mutations. In this study, the authors evaluated the incidence of LKB1 loss of expression in surgical tissues from NSCLC patients to determine possible expression patterns and characteristics of the LKB1-less phenotype and to find potential disease-related associations. They selected 248 consecutive patients with operable stage I-IIIA NSCLC who underwent surgery. Co-mutations with LKB1 gene loss are also examined for occurrence and characteristics. The article presents a detailed analysis of LKB loss correlation with other genes in NSCLC. However, I have a few concerns.

1. What criteria were used to include or exclude patients' tissues during the selection process? It is necessary to include this information in the methods section. 

2. Despite this, the authors selected 248 stage I-IIIA NSCLC patients who received curative surgery between 2015 and 2023. To compare LKB1 loss of expression studies, authors should also consider normal tissues in each experiment. 

3. Have the patients undergoing chemotherapy or other therapies, for example, adjuvant or neoadjuvant? It should be noted in the text.  

4. The authors presented images of nuclear and cytoplasmic staining. However, because the sizes of the samples vary, the results should be displayed in a box plot for both nuclear and cytoplasmic membrane expression.

5. What was the scale bar used for the IHC images?

Comments on the Quality of English Language

Extensive editing of English language required.

Author Response

1/05/2024

Dear Editor,

Thank you very much for taking the time to review this manuscript.

Please find the detailed responses below and the corresponding revisions highlighted in track changes in the re-submitted files

Please find here below answers one by one.

We are looking forward to hearing from you.

Best regards

Eleni Lagoudaki

  1. What criteria were used to include or exclude patients' tissues during the selection process? It is necessary to include this information in the methods section.

We agree with this comment. Therefore, we have added the following:

To perform this retrospective translational research study we retrieved from the records of the Pathology department of the University General Hospital of Heraklion the pathololgy reports of patients who underwent surgery with curative intent for NSCLC between 2007 and 2023, and after re-evaluating each patient’s slides of excised tumor tissue and lymph node specimens i) for histologic subtype according 2015 WHO classification [39] and ii) stage in accordance to the 8th American Joint Committee on Cancer Staging AJCC [40], we enrolled 188 treatment naïve patients with operable, stage I-IIIA, NSCLC harboring pathologically confirmed metastasis to regional and distant lymph nodes, as the experimental group as well as 60, also treatment naïve, patients with pathologically confirmed tumor free lymph nodes, as control group. Patients selection was irrespective of the histological type and mutational status of the tumor so to include the entire repertoire of histologies of NSCLCs and all the potential mutational cohorts.

  1. Despite this, the authors selected 248 stage I-IIIA NSCLC patients who received curative surgery between 2015 and 2023. To compare LKB1 loss of expression studies, authors should also consider normal tissues in each experiment.

Thank you for pointing this out. We did consider the use of normal tissue in our experiments. Along with tumor tissue, matched adjacent normal lung tissue, at least 5 cm away from the tumor lesions was selected and evaluated in all experiments. Regarding LKB1 immunohistochemical expression, control lung tissue showed uniform LKB1 expression. LKB1 protein was present in all the cells of the epithelia of the trachea and bronchus, except goblet cells, whereas it was almost undetectable in the alveolar pneumocytes. 

Additionally, as commented in Supplementary S2, ciliated bronchial epithelium adjacent to tumor served as internal positive control of LKB1 protein loss of expression. Indeed, LKB1was highly expressed in the apical surface (i.e. cilia), consistent with LKB1 known roles in the establishment and maintenance of epithelial polarity.

  1. Have the patients undergoing chemotherapy or other therapies, adjuvant or neoadjuvant ? It should be noted in the text.

We agree with this comment. The NSCLCs tumors included in our study were from   treatment naïve patients. None of the patients had received neoadjuvant chemotherapy, radiotherapy or immunotherapy prior to surgery.

  1. The authors presented images of nuclear and cytoplasmic staining. However because the sizes of the samples vary, the results should be displayed in a box plot for both nuclear and cytoplasmic membrane expression.

Thanks for this comment. Our aim was to include in our study all histological subtypes of NSCLC, even the rarest -undifferentiated- ones (Pleomorphic spindle cell carcinomas, Pleomorphic large cell carcinomas,  Adenosquamous carcinomas, Large cell carcinomas), which so far have not been studied for the status of LKB1 expression. 

The indeed significant difference in sample size is due to the different incidence of the different histological subtypes.

In our study, immunohistochemical screening has been performed for the expression of 16 different proteins, with distinctly different localizations (cytoplasmic, nuclear and membrane), according to their function as detailed in Supplementary table 2 and has been assessed in various ways on the basis of published, comprehensive literature data. Regardless of the assessment mode, all laboratory variables have been transformed into binary data for the purpose of the statistical analysis. Binary data cannot be represented in boxplots as they are not applicable for binary or ordered data.

Furthermore the expression status of all proteins examined in all different histological subtypes is presented extensively in Supplementary table S4.

  1. What was the scale bar used for the IHC images ?

Thank you for pointing this out. The scale bar used was 50 μm.

We have, accordingly, revised the photos. This is now inserted in all photos.

Reviewer 2 Report

Comments and Suggestions for Authors

The original article presents interesting findings and the topic is really important from clinical point of view. A lot of experimental work was done.  248 consecutive surgical specimens from stage I-IV NSCLC patients, were analyzed for the expression of LKB1 and pAMPK protein, for antitumor immunity response regulators STING and PD-L1, for pro-angiogenic, EMT and cell cycle target as well as metastasis related molecules (VEGFC, PDGFRα, PDGFRβ, p53, p16, Cyclin D1, ZEB1, CD24); by immunohistochemistry, for LKB1 and NEDD9 RNA levels by PCR as well as for 2 KRAS exon 2 and BRAFV600E mutations by Sanger sequencing. The results are valuable and worth to publish in Cancers. Authors should check minor spelling throughout the article (for example - line 647).

Author Response

Reply to reviewer 2

Dear Editor,

Thank you very much for taking the time to review this manuscript and for pointing out this: “Authors should check minor spelling throughout the article (for example - line 647”.

We have now checked the manuscript for minor spelling. No, we did check minor spelling for line 647 and we have revised as following: 

Our results are in accordance with data from a recent paper investigating the contribution of four missense LKB1 somatic mutations in the biology of the tumor that showed that LKB1R87K and to a lesser extent LKB1D194Y mutant isoform, induced cell motility, promoted cytoskeleton regulation, along with β-catenin degradation and loss at the cell junctions in the 3D cultures [15].

We are looking forward to hearing from you.

Yours sincerely

Eleni Lagoudaki
